# Spherical convolutions and their application in molecular modelling

**Wouter Boomsma**
Department of Computer Science
University of Copenhagen
wb@di.ku.dk

**Jes Frellsen**
Department of Computer Science
IT University of Copenhagen
jefr@itu.dk

## Abstract

Convolutional neural networks are increasingly used outside the domain of image analysis, in particular in various areas of the natural sciences concerned with spatial data. Such networks often work out-of-the box, and in some cases entire model architectures from image analysis can be carried over to other problem domains almost unaltered. Unfortunately, this convenience does not trivially extend to data in non-euclidean spaces, such as spherical data. In this paper, we introduce two strategies for conducting convolutions on the sphere, using either a spherical-polar grid or a grid based on the cubed-sphere representation. We investigate the challenges that arise in this setting, and extend our discussion to include scenarios of spherical volumes, with several strategies for parameterizing the radial dimension. As a proof of concept, we conclude with an assessment of the performance of spherical convolutions in the context of molecular modelling, by considering structural environments within proteins. We show that the models are capable of learning non-trivial functions in these molecular environments, and that our spherical convolutions generally outperform standard 3D convolutions in this setting. In particular, despite the lack of any domain specific feature-engineering, we demonstrate performance comparable to state-of-the-art methods in the field, which build on decades of domain-specific knowledge.

## 1  Introduction

Given the transformational role that convolutional neural networks (CNNs) have had in the area of image analysis, it is natural to consider whether such networks can be efficiently applied in other contexts. In particular spatially embedded data can often be interpreted as images, allowing for direct transfer of neural network architectures to these domains. Recent years have demonstrated interesting examples in a broad selection of the natural sciences, ranging from physics (Aurisano et al., 2016; Mills et al., 2017) to biology (Wang et al., 2016; Min et al., 2017), in many cases showing convolutional neural networks to substantially outperform existing methods.

The standard convolutional neural network can be applied naturally to data embedded in a Euclidean space, where uniformly spaced grids can be trivially defined. For other manifolds, such as the sphere, it is less obvious, and to our knowledge, convolutional neural networks for such manifolds have not been systematically investigated. In particular for the sphere, the topic has direct applications in a range of scientific disciplines, such as the earth sciences, astronomy, and modelling of molecular structure.

This paper presents two strategies for creating *spherical convolutions*, as understood in the context of convolutional neural networks (i.e., discrete, and efficiently implementable as tensor operations). The first is a straightforward periodically wrapped convolution on a spherical-polar grid. The second builds on the concept of a cubed-sphere (Ronchi et al., 1996). We proceed with extending these

strategies to include the radial component, using concentric grids, which allows us to conduct convolutions in spherical volumes.

Our hypothesis is that these concentric spherical convolutions should outperform standard 3D convolutions in cases where data is naturally parameterized in terms of a radial component. We test this hypothesis in the context of molecular modelling. We will consider structural environments in a molecule as being defined from the viewpoint of a single amino acid or nucleotide: how does such an entity experience its environment in terms of the mass and charge of surrounding atoms? We show that a standard convolutional neural network architectures can be used to learn various features of molecular structure, and that our spherical convolutions indeed outperform standard 3D convolutions for this purpose. We conclude by demonstrating state-of-the art performance in predicting mutation induced changes in protein stability.

## 2 Spherical convolutions

Conventional CNNs work on discretized input data on a grid in $\mathbb{R}^n$, such as time series data in $\mathbb{R}$ and image data in $\mathbb{R}^2$. At each convolutional layer $l$ a CNN performs discrete convolutions (or a correlation)

$$[f * k^i](\mathbf{x}) = \sum_{\mathbf{x}' \in \mathbb{Z}^n} \sum_{c=1}^{C_l} f_c(\mathbf{x}') k_c^i(\mathbf{x} - \mathbf{x}') \tag{1}$$

of the input feature map $f : \mathbb{Z}^n \to \mathbb{R}^{C_l}$ and a set of $C_{l+1}$ filters $k^i : \mathbb{Z}^n \to \mathbb{R}^{C_l}$ (Cohen and Welling, 2016; Goodfellow et al., 2016). While such convolutions are equivariant to translation on the grid, they are not equivariant to scaling (Cohen and Welling, 2016). This means that in order to preserve the translation equivariance in $\mathbb{R}^n$, conventional CNNs rely on the grid being uniformly spaced within each dimension of $\mathbb{R}^n$. Constructing such a grid is straightforward in $\mathbb{R}^n$. However, for convolutions on other manifolds such as the 2D sphere, $\mathbb{S}^2 = \{\mathbf{v} \in \mathbb{R}^3 | \mathbf{v}\mathbf{v}^\mathsf{T} = 1\}$, no such planar uniform grid is available, due to the non-linearity of the space (Mardia and Jupp, 2009). In this section, we briefly discuss the consequences of using convolutions in the standard non-uniform spherical-polar grid, and present an alternative grid for which the non-uniformity is expected to be less severe.

### 2.1 Convolutions of features on $\mathbb{S}^2$

A natural approach to a discretization on the sphere is to represent points $\mathbf{v}$ on the sphere by their spherical-polar coordinates $(\theta, \phi)$ and construct uniformly spaced grid in these coordinates, where the spherical coordinates are defined by $\mathbf{v} = (\cos\theta, \sin\theta\cos\phi, \sin\theta\sin\phi)^\mathsf{T}$. Convolutions on such a grid can be implemented efficiently using standard 2D convolutions when taking care of using periodic padding at the $\phi$ boundaries. The problem with a spherical-polar coordinate grid is that it is highly non-equidistant when projected onto the sphere: the distance between grid points becomes increasingly small as we move from the equator to the poles (figure 1, left). This reduces the ability to share filters between different areas of the sphere.

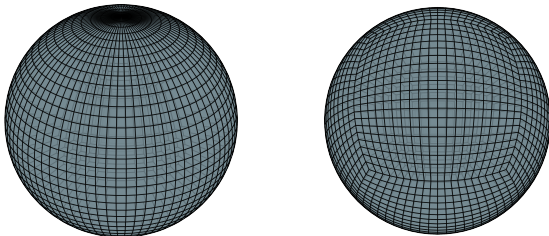

Figure 1: Two realizations of a grid on the sphere. Left: a grid using equiangular spacing in a standard spherical-polar coordinate system, and Right: An equiangular cubed-sphere representation, as described in Ronchi et al. (1996).

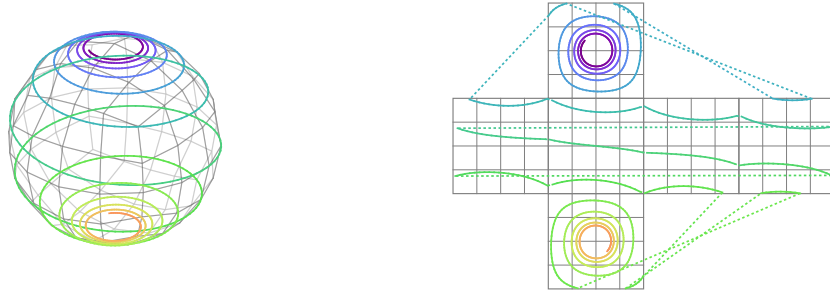

Figure 2: Left: A cubed-sphere grid and a curve on the sphere. Right: The six planes of a cubed-sphere representation and the transformation of the curve to this representation.

As a potential improvement, we will investigate a spherical convolution based on the *cubed-sphere transformation* (Ronchi et al., 1996). The transformation is constructed by decomposing the sphere into six patches defined by projecting the circumscribed cube onto the sphere (figure 1, right). In this transformation a point on the sphere $\mathbf{v} \in \mathbb{S}^2$ is mapped to a patch $b \in \{1, 2, 3, 4, 5, 6\}$ and two coordinates $(\xi, \eta) \in [-\pi/4, \pi/4[^2$ on that patch. The coordinate are given by the angles between the axis pointing to the patch and $\mathbf{v}$ measured in the two coordinate planes perpendicular to the patch. For instance the vectors $\{\mathbf{v} \in \mathbb{S}^2 | v_x > v_y \text{ and } v_x > v_z\}$ map to patch $b = 1$ and we have $\tan \xi = v_y/v_x$ and $\tan \eta = v_z/v_x$. The remaining mappings are described by Ronchi et al. (1996).

If we grid the two angles $(\xi, \eta)$ uniformly in the cubed-sphere transformation and project this grid onto the sphere, we obtain a grid that is more regular (Ronchi et al., 1996), although it has artefacts in the 8 corners of the circumscribed cube (figure 1, right). The *cubed-sphere convolution* is then constructed by applying the conventional convolution in equation (1) to a uniformly spaced grid on each of the six cubed shape patches. This construction has two main advances: 1) within each patch, the convolution is almost equivariant to translation in $\xi$ and $\eta$ and 2) features on the cubed-sphere grid can naturally be expressed using tensors, which means that the spherical convolution can be efficiently implemented on a GPU. When implementing convolutions and pooling operations for the cubed-sphere grid, one has to be careful in padding each patch with the contents of the four neighbouring patches, in order to preserve the wrapped topology of the sphere (figure 2, right).

Both of these two approaches to spherical convolutions are hampered by a lack of rotational equivariance, which restricts the degree with which filters can be shared over the surface of the sphere, leading to suboptimal efficiency in the learning of the parameters. Despite this limitation, for capturing patterns in spherical volumes, we expect that the ability to express patterns naturally in terms of radial and angular dimensions has advantages over standard 3D convolutions. We test this hypothesis in the following sections.

## 2.2 Convolutions of features on $\mathbb{B}^3$

The two representations from figure 1 generalize to the ball $\mathbb{B}^3$ by considering concentric shells at uniformly separated radii. In the case of the cubed-sphere, this means that a vector $\mathbf{v} \in \mathbb{B}^3$ is mapped to the unique coordinates $(r, b, \xi, \eta)$, where $r = \sqrt{\mathbf{v}\mathbf{v}^{\mathsf{T}}}$ is the radius and $(b, \xi, \eta)$ are the cubed-sphere coordinates at $r$, and we construct a uniform grid in $r$, $\xi$ and $\eta$. Likewise, in the spherical-polar case, we construct a uniform grid in $r$, $\theta$ and $\phi$. We will refer to these grids as *concentric cubed-sphere grid* and *concentric spherical-polar grid* respectively (figure 3). As is the case for their $\mathbb{S}^2$ counterparts, features on these grids can be naturally expressed using tensors.

We can apply the conventional 3D convolutions in equation (1) to features on the concentric cubed-sphere and the concentric spherical-polar grids, and denote these as *concentric cubed-sphere convolution* (CCSconv) and *concentric spherical-polar convolution* (CSPconv). For fixed $r$, the convolutions will thus have the same properties as in the $\mathbb{S}^2$ case. In these concentric variants, the convolutions will not be equivariant to translations in $r$, which again reduces the potential to share filter parameters.



Figure 3: Three realizations of a grid on the ball. Left: a grid using equiangular spacing in a standard spherical-polar coordinate system (concentric spherical-polar grid). Center: An equiangular cubed-sphere representation, as described in Ronchi et al. (1996) (concentric cubed-sphere grid). Right: a Cartesian grid.

We propose to address this issue in three ways. First, we can simply apply the convolution over the full range of $r$ with a large number of filters $C_{l+1}$ and hope that the network will automatically allocate different filters at different radii. Secondly, we can make the filters $k^i(\mathbf{x} - \mathbf{x}', x_r)$ depend on $r$, which corresponds to using different (possibly overlapping) filters on each spherical shell (*conv-banded-disjoint*). Thirdly, we can divide the $r$-grid into segments and apply the same filter within each segment (*conv-banded*), potentially with overlapping regions (depending on the stride). The three approaches are illustrated in figure 4.

In the experiments below, we will be comparing the performance of our concentric spherical convolution methods to that of a simple 3D convolution in a Cartesian grid (figure 3, right).

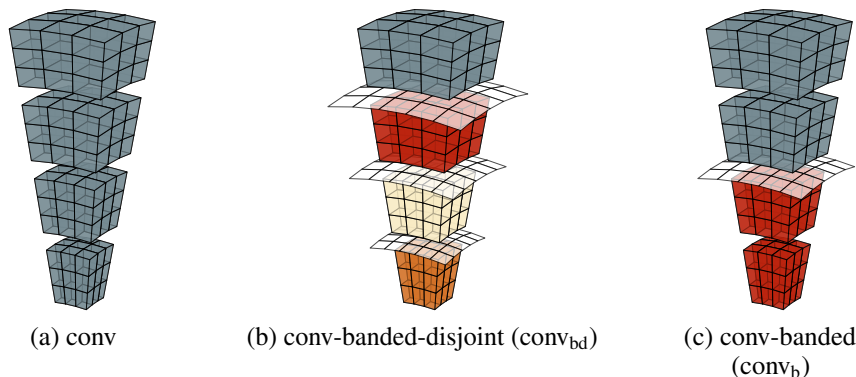

(a) conv        (b) conv-banded-disjoint ($conv_{bd}$)        (c) conv-banded ($conv_b$)

Figure 4: Three strategies for the radial component of concentric cubed-sphere or concentric spherical convolutions. (a) conv: The same convolution-filter is applied to all values of $r$, (b) conv-banded-disjoint ($conv_{bd}$): convolution-filters are only applied in the angular directions, using different filters for each block in $r$, (c) conv-banded ($conv_b$): convolutions are applied within radial segments, Note that for visual clarity, we use a stride of 3 in this figure, although we use a stride of 1 in practice.

## 3 Modelling structural environments in molecules

In the last decades, substantial progress has been made in the ability to simulate and analyse molecular structures on a computer. Much of this progress can be ascribed to the molecular force fields used to capture the physical interactions between atoms. The basic functional forms of these models were established in the late 1960s, and through gradual refinements they have become a success story of molecular modelling. Despite these positive developments, the accuracy of molecular force fields is known to still be a limiting factor for many biological and pharmaceutical applications, and further improvements are necessary in this area to increase the robustness of methods for e.g. protein prediction and design. There are indications that Machine Learning could provide solutions to such challenges. While, traditionally, most of the attention in the Machine Learning community has been dedicated

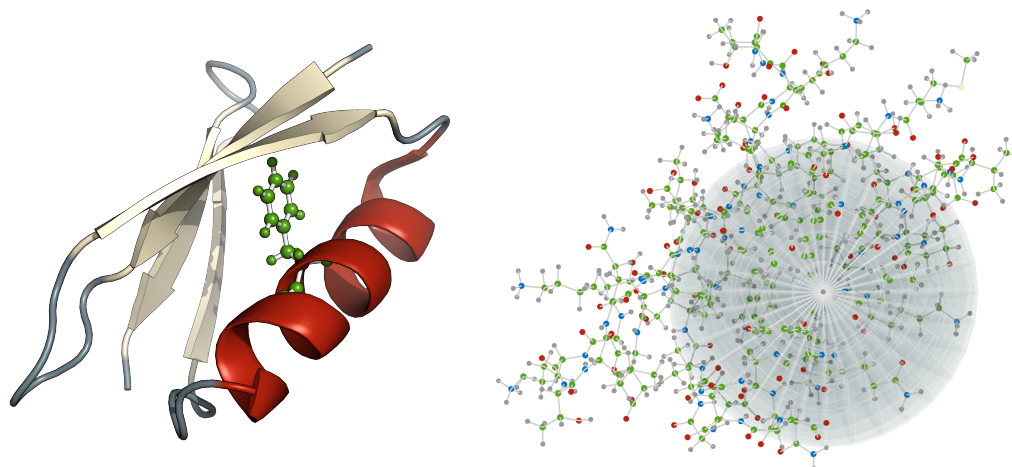

Figure 5: Example of the environment surrounding an amino acid in a protein, in this case the phenylalanine at position 30 in protein GB1 (PDB ID: 2GB1). Left: a cartoon representation of GB1, where the helix is red, the sheets are yellow and the coils are grey. The phenylalanine is shown using an atomic representation in green. Right: an atomic representation of GB1, where carbon atoms are green, oxygen atoms are red, nitrogen atoms are blue and hydrogen atoms are grey. A sphere centered at the $C_\alpha$ atom of the phenylalanine with a radius of 12Å is shown in grey.

to predicting structural features from amino acid sequences (e.g. secondary structure, disorder, and contact prediction), there are increasingly applications taking three dimensional molecular structure as input (Behler and Parrinello, 2007; Jasrasaria et al., 2016; Schütt et al., 2017; Smith et al., 2017). In particular in the field of quantum chemistry, a number of studies have demonstrated the ability of deep learning techniques to accurately predict energies of molecular systems. Common to many of these methods is a focus on manually engineered features, where the molecular input structure is encoded based on prior domain-specific knowledge, such as specific functional relationships between atoms and their environments (Behler and Parrinello, 2007; Smith et al., 2017). Recently, a few studies have demonstrated the potential of automatically learning such features, by encoding the molecular structural input in a more domain-agnostic manner, for instance considering only pairwise distance matrices (Schütt et al., 2017), space filling curves (Jasrasaria et al., 2016), or basic structural features (Wallach et al., 2015).

The fact that atomic forces are predominantly distance-based suggests that molecular environments are most naturally represented with a radial-based parameterization, which makes it an obvious test case for the convolutions presented in the previous section. If successful, such convolutions could allow us to make inferences directly from the raw molecular structure of a molecule, avoiding the need of manual feature engineering. We will consider the environments that each amino acids experience within its globular protein structure as images in the 3-ball. Figure 5 shows an example of the environment experienced by an arbitrarily chosen amino acid in the GB1 protein (PDB ID: 2GB1). Although distorted by the fish-eye perspective, the local environment (right) displays several key features of the data: we see clear patterns among neighboring atoms, depending on their local structure, and we can imagine the model learning to recognize hydrogen bonds and charge interactions between an amino acid and its surroundings.

Our representation of the molecular environment includes all atoms within a 12 Å radius of the $C_\alpha$ atom of the amino acid in question. Each atom is represented by three fundamental properties: 1) its position relative to the amino acid in question (i.e., the position in the grid), 2) its mass, and 3) its partial charge, as defined by the amber99sb force field (Hornak et al., 2006). We construct two types of models, which are identical except for their output. The first outputs the propensity for different secondary structure labels at a given position (i.e., helix, extended, coil), while the second outputs the propensity for different amino acid types. Each of these models will be implemented with both the Cartesian, the concentric spherical and the concentric cubed-sphere convolutions. Furthermore, for the concentric cubed-sphere convolutions, we compare the three strategies for dealing with the radial component illustrated in figure 4.

Table 1: The architecture of the CNN where $o$ represent the output size, which is 3 for secondary structure output and 20 for amino acid output. As an example, we use the convolutional filter sizes from the concentric cubed-sphere (CCS) case. Similar sizes are used for the other representations.

| Layer | Operation | Filter / weight size | Layer output size |
|---|---|---|---|
| 0 | Input | | $6 \times 24 \times 38 \times 38 \times 2$ |
| 1 | CCSconv + ReLU | $3 \times 5 \times 5 \times 2 \times 16$ | $6 \times 22 \times 19 \times 19 \times 16$ |
| 1 | CCSsumpool | $1 \times 3 \times 3$ | $6 \times 22 \times 10 \times 10 \times 16$ |
| 2 | CCSconv + ReLU | $3 \times 3 \times 3 \times 16 \times 32$ | $6 \times 20 \times 10 \times 10 \times 32$ |
| 2 | CCSsumpool | $3 \times 3 \times 3$ | $6 \times 9 \times 5 \times 5 \times 32$ |
| 3 | CCSconv + ReLU | $3 \times 3 \times 3 \times 32 \times 64$ | $6 \times 7 \times 5 \times 5 \times 64$ |
| 3 | CCSsumpool | $1 \times 3 \times 3$ | $6 \times 7 \times 3 \times 3 \times 64$ |
| 4 | CCSconv + ReLU | $3 \times 3 \times 3 \times 64 \times 128$ | $6 \times 5 \times 3 \times 3 \times 128$ |
| 4 | CCSsumpool | $1 \times 3 \times 3$ | $6 \times 5 \times 3 \times 3 \times 128$ |
| 5 | Dense + ReLU | $34\,560 \times 2\,048$ | $2\,048$ |
| 6 | Dense + ReLU | $2\,048 \times 2\,048$ | $2\,048$ |
| 7 | Dense + Softmax | $2\,048 \times o$ | $o$ |

## 3.1 Model architecture

The input to the network is a grid (concentric cubed-sphere, concentric spherical polar or Cartesian). Each voxel has two input channels: the mass of the atom that lies in the given bin and the atom's partial charge (or zeros if no atom is found). The resolution of the grids are chosen so that the maximum distance within a bin is 0.5Å, which ensures that bins are occupied by at most one atom. The radius of the ball is set to 12Å, since most physical interactions between atoms occur within this distance (Irbäck and Mohanty, 2006). This gives us an input tensor of shape $(b = 6, r = 24, \xi = 38, \eta = 38, C_1 = 2)$ for the concentric cubed-sphere case, $(r = 24, \theta = 76, \phi = 151, C_1 = 2)$ for the concentric spherical polar case, and $(x = 60, y = 60, z = 60, C_1 = 2)$ for the Cartesian case.

We use a deep model architecture that is loosely inspired by the VGG models (Simonyan and Zisserman, 2015), but employs the convolution operators described above. Our models have four convolutional layers followed by three dense layers, as illustrated in table 1. Each convolutional layer is followed by rectified linear unit (ReLU) activation function (Hahnloser et al., 2000; Glorot et al., 2011) and a sum pooling operation which is appropriately wrapped in the case of the concentric cubed-sphere and the concentric spherical polar grid. We use sum pooling since the input features, mass and partial charge, are both physical quantities that are naturally additive. The total number of parameters is the models (with the amino acid output) are $75\,313\,253$ (concentric cubed-sphere), $69\,996\,645$ (concentric spherical polar), and $61\,159\,077$ (Cartesian). Furthermore, for the concentric cubed-sphere case, we include a comparison of the two alternative strategies for the radial component: the $\text{conv}_b$ and the $\text{conv}_{bd}$, which have $75\,745\,333$ and $76\,844\,661$ parameters respectively. Finally, to see the effect of convolutions over a purely dense model, we include a baseline model where the convolutional layers are replaced with dense layers, but otherwise following the same architecture, and roughly the same number of parameters ($66\,670\,613$).

## 3.2 Training

We minimized the cross-entropy loss using Adam (Kingma and Ba, 2015), regularized by penalizing the loss with the sum of the $L_2$ of all weights, using a multiplicative factor of $0.001$. All dense layers also used dropout regularization with a probability of $0.5$ of keeping a neuron. The models were trained on NVIDIA Titan X (Pascal) GPUs, using a batch size of $100$ and a learning rate of $0.0001$.

The models were trained on data set of high resolution crystal structures. A large initial (non-homology-reduced) data set was constructed using the PISCES server (Wang and Dunbrack, 2003). For all structures, hydrogen atoms were added using the Reduce program (Word et al., 1999), after which partial charges were assigned using the OpenMM framework (Eastman et al., 2012), using the amber99sb force field (Hornak et al., 2006). During these stages strict filters were applied to remove structures that 1) were incomplete (missing chains or missing residues compared to the seqres

entry), 2) had chain breaks, 3) failed to parse in OpenMM, or 4) led the Reduce program to crash. Finally, the remaining set was resubmitted to the PISCES server, where homology-reduction was done at the 30% level. This left us with 2336 proteins, out of which 1742 were used for training, 10 for validation, and the remainder was set aside for testing. The homology-reduction ensures that any pair of sequences in the data set are at most 30% identical at the amino-acid-level, which allows us to safely split the data into non-overlapping sets.

# 4 Results

We now discuss results obtained with the secondary structure and amino acid models, respectively. Despite the apparent similarity of the two models, the two tasks have substantially different biological implications: secondary structure is related to the 3D structure locally at a given position in a protein, i.e. whether the protein assumes a helical or a more extended shape. In contrast, amino acid propensities describe allowed mutations in a protein, which is related to the fundamental biochemistry of the molecule, and is relevant for understanding genetic disease and for design of new proteins.

## 4.1 Learning the DSSP secondary structure function

Predicting the secondary structure of a protein conditioned on knowledge of the three dimensional structure is not considered a hard problem. We include it here because we are interested in the ability of the neural network to *learn* the function that is typically used to annotate three dimensional structures with secondary structure, in our case DSSP (Kabsch and Sander, 1983). Interestingly, the different concentric convolutional models are seen to perform about equally well on this problem (table 2, Q3), marginally outperforming the Cartesian convolution and substantially outperforming the dense baseline model.

To get a sense of the absolute performance, we would ideally compare to existing methods on the same problem. However, rediscovering the DSSP function is not a common task in bioinformatics, and not many tools are available that would constitute a meaningful comparison, in particular because secondary structure annotation algorithms use different definitions of secondary structure. We here use the TORUSDBN model (Boomsma et al., 2008, 2014) to provide such a baseline. The model is sequential in the sequence of a protein, and thus captures local structural information only. While the model is originally designed to sample backbone dihedral angles conditioned on an amino acid sequence or secondary structure sequence, it is generative, and can thus be used in reverse and provide the most probable secondary structure or amino acid sequence given using viterbi decoding. Most importantly, it is trained on DSSP, making it useful as a comparison for this study. Included as the last row in table 2, TORUSDBN demonstrates slightly lower performance compared to our convolutional approaches, illustrating that most of the secondary structure signal is encoded in the local angular preferences. It is encouraging to see that the convolutional networks capture all these local signals, but obtain additional performance through more non-local interactions.

### 4.1.1 Learning amino acid propensities

Compared to secondary structure, predicting the amino acid propensity is substantially harder—partly because of the larger sample space, but also because we expect such preferences to be defined by more global interaction patterns. Interestingly, the two concentric convolutions perform about equally well, suggesting that the added regularity of the cubed-sphere representation does not provide a substantial benefit for this case (table 2, Q20). However, both methods substantially outperform the standard 3D convolution, which again outperforms the dense baseline model. We also note that there is now a significant difference between the three radial strategies, with conv-banded-disjoint (bd) and conv-banded (b) both performing worse than the simpler case of using a single convolution over the entire $r$-range. Again, we include TorusDBN as an external reference. The substantially lower performance of this model confirms that the amino acid label prediction task depends predominantly on non-local features not captured by this model. Finally, we include another baseline: the most frequent amino acid observed at this position among homologous (evolutionarily related) proteins. It is remarkable that the concentric models (which are trained on a homology-reduced protein set), are capable of learning the structural preferences of amino acids to the same extent as the information that is encoded as genetic variation in the sequence databases. This strongly suggests the ability of our models to learn general relationships between structure and sequence.

Table 2: Performance of various models in the prediction of (a) DSSP-style secondary structure conditioned and (b) amino acid propensity conditioned on the structure. The Q3 score is defined as the percentage of correct predictions for the three possible labels: helix, extended and coil. The Q20 score is defined as the percentage of correct predictions for the 20 possible amino acid labels.

| Model | Q3 (secondary structure) | Q20 (amino acid) |
|---|---|---|
| CCSconv | **0.933** | **0.564** |
| CCSconv$_{bd}$ | 0.931 | 0.515 |
| CCSconv$_b$ | 0.932 | 0.548 |
| CSPconv | 0.932 | 0.560 |
| Cartesian | 0.922 | 0.500 |
| CCSdense | 0.888 | 0.348 |
| PSSM | - | 0.547 |
| TORUSDBN | 0.894 | 0.183 |

### 4.1.2 Predicting change-of-stability

The models in the previous section not only predict the most likely amino acid, but also the entire distribution. A natural question is whether the ratio of probabilities of two amino acids according to this distribution is related to the change of stability induced by the corresponding mutation. We briefly explore this question here.

The stability of a protein is the difference in free energy $\Delta G$ between the folded and unfolded conformation of a protein. The *change* in stability that occurs as a consequence of a mutation is thus frequently referred to as $\Delta\Delta G$. These values can be measured experimentally, and several data sets with these values are publicly available. As a simple approximation, we can interpret the sum of negative log-probabilities of each amino acid along the sequence as a free energy of the folded state $G_f$. To account for the free energy of the unfolded state, $G_u$, we could consider the negative log-probability that the amino acid in question occurs in the given amino acid sequence (without conditioning on the environment). Again, assuming independence between sites in the chain, this could be modelled by simply calculating the log-frequencies of the different amino acids across the data set, and summing over all sites of the specific protein to get the total free energy. Subtracting these two pairs of values for the wild type (W) and mutant (M) would give us a rough estimate of the $\Delta\Delta G$, which due to our assumption of independence between sites simplifies to just the difference in values at the given site:

$$\Delta\Delta G(\bar{W},\bar{M}) = (G_f(M_n) - G_u(M_n)) - (G_f(W_n) - G_u(W_n)), \qquad (2)$$

where $\bar{W}$ and $\bar{M}$ denote the full wild type and mutant sequence respectively, and $W_n$ and $M_n$ denote the amino acids of wild type and mutant at the site $n$ at which they differ. Given the extensive set of simplifying assumptions in the argument above, we do not use the expression in equation (2) directly but rather use the four log-probabilities ($G_f(M_n), G_u(M_n), G_f(W_n), G_u(W_n)$) as input to a simple regression model (a single hidden layer neural network with 10 hidden nodes and a ReLU activation function), trained on experimentally observed $\Delta\Delta G$ data. We calculate the performance on several standard experimental data sets on mutation-induced change-of-stability, in each case using 5-fold cross validation, and reporting the correlation between experimentally measured and our calculated $\Delta\Delta G$. As a baseline, we compare our performance to two of the best known programs for calculating $\Delta\Delta G$: Rosetta and FoldX. The former were taken from a recent publication (Conchúir et al., 2015), while the latter were calculated using the FoldX program (version 4). The comparison shows that even a very simple approach based on our convolutional models produces results that are comparable to the state-of-the-art in the field (table 3). This is despite the fact that we use a rather crude approximation of free energy, and that our approach disregards the fact that a mutation at a given site modifies the environment grids of all amino acids within the 12 Å range. Although these initial results should therefore not be considered conclusive, they suggest that models like the ones we propose could play a future role in $\Delta\Delta G$ predictions.

Apart from the overall levels of performance, the most remarkable feature of table 3 is that it shows equal performance for the Cartesian and concentric cubed-sphere convolutions, despite the fact that the former displayed substantially lower Q20 scores. This peculiar result points to an interesting

Table 3: Pearson correlation coefficients between experimentally measured and predicted changes of stability for several sets of published stability measurements.

|  | Rosetta | FoldX | CCSconv | CSPconv | Cartesian |
|---|---|---|---|---|---|
| Kellogg | 0.65 | 0.70 | 0.66 | 0.64 | 0.66 |
| Guerois | 0.65 | 0.73 | 0.66 | 0.64 | 0.66 |
| Potapov | 0.52 | 0.59 | 0.52 | 0.51 | 0.52 |
| ProTherm* | 0.44 | 0.53 | 0.49 | 0.48 | 0.49 |

caveat in the interpretation of the predicted distribution over amino acids for a given environment. At sufficiently high resolution of the structural environment, a perfect model would be able to reliably predict the identity of the wild type amino acid by the specific shape of the hole it left behind. This means that as models improve, the entropy of the predicted amino acid distributions is expected to decrease, with increasingly peaked distributions centered at the wild type. An increased sensitivity towards the exact molecular environment will therefore eventually decrease the models ability to consider other amino acids at that position, leading to lower $\Delta\Delta G$ performance. The missing ingredient in our approach is the structural rearrangement in the environments that occurs as a consequence of the mutation. A full treatment of the problem should average the predictions over the available structural variation, and structural resampling is indeed part of both Rosetta and FoldX. For these reasons, it is difficult to make clear interpretations of the relative differences in performance of the three convolution procedures in table 3. The overall performance of all three, however, indicates that convolutions might be useful as part of a more comprehensive modelling strategy such as those used in Rosetta and FoldX.

## 5 Conclusions

Convolutional neural networks are a powerful tool for analyzing spatial data. In this paper, we investigated the possibility of extending the applicability of the technique to data in the 3-ball, presenting two strategies for conducting convolutions in these spherical volumes. We assessed the performance of the two strategies (and variants thereof) on various tasks in molecular modelling, and demonstrate a substantial potential of these such concentric convolutional approaches to outperform standard 3D convolutions for such data.

We expect that further improvements to the concentric convolution approach can be obtained by improving the spherical convolutions themselves. In particular, a convolution operation that is rotationally equivariant would provide greater data efficiency than the approach used here. Very recently, a procedure for conducting convolutions in SO(3) was proposed, which seems to provide an elegant solution to this problem (Cohen et al., 2018).

Finally, we note that while this manuscript was in review, another paper on the application of convolutional neural networks for predicting amino acid preferences conditioned on structural environments was published, by Torng and Altman (Torng and Altman, 2017). Their study is conceptually similar to one of the applications described in this paper, but uses a Cartesian grid and standard 3D convolution (in addition to other minor differences, such as a one-hot atom type encoding). While Torng and Altman present a more thorough biological analysis in their paper than we do here, the accuracy they report is considerably lower than what we obtained. Based on the comparisons reported here, we anticipate that models such as theirs could be improved by switching to a concentric representation.

## 6 Availability

The spherical convolution Tensorflow code and the datasets used in this paper are available at `https://github.com/deepfold`.

**Acknowledgments**

This work was supported by the Villum Foundation (W.B., grant number VKR023445).

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
