[Reviews · NeurIPS 2017]

Reviewer 1



The authors formulate a spherical convolution operation for features on a sphere that can be efficiently implemented using tensor operations on a GPU. This operation is evaluated as part of deep learning models for molecular modeling. The authors demonstrate that CNNs with spherical convolutions outperform fully connected models on two tasks in that domain and that three flavors of the spherical convolution operation results in similar performance. However, a more interesting baseline than a fully connected model would be a CNN with standard convolution operations. In particular, what would happen if the molecules were parameterized in the euclidean space and modeled with conventional CNNs? It is not clear if spherical convolutions would outperform standard convolutions on these tasks. There are several existing models that successfully utilize conventional 3D convolutions for molecular modeling (AtomNet, for example). While this proof of concept is intriguing, without demonstrating the added value of spherical convolutions as compared to regular convolutions, it is not clear that this extension could be useful in practice.

Reviewer 2



Paper summary: The authors propose to separate the surface of a sphere using the equiangular cubed sphere representation of Ronchi et al. (1996) in order to do a spherical convolution. This convolution has the advantage of being almost invariant to rotation of the volume and should have application in earth science, astronomy and in molecular modeling. The authors demonstrate the usefulness of their approach on practical tasks of this later field. Strengths and weaknesses: The paper is clearly written and easy to read. Also, I appreciate the amount of work that has been put in this paper as molecular modeling is a field with a steep learning curve, where the data is hard to pre-process, and where clear prediction task is sometime hard to define. Although I appreciate the importance of spherical convolution on spherical data and the comparison with a purely dense model, there are other comparisons I believe would be of interest. Namely, I would like to know if empirical comparison with 3D convolution (ex: https://keras.io/layers/convolutional/#conv3d) in a cube is possible. For example, in the case of molecular modeling, the cube could be centered on the atom of interest. Also, it would also be of interest to compare the spherical convolution using different coordinate system: the equiangular cubed sphere (as proposed, Figure 1b) and the standard equiangular spacing (Figure 1a). Finally, it would be nice if the conclusion contained ways to improve the proposed approach. Quality: I found the paper well written and easy to understand. I found few minor typos that are highlighted at the end of this review. Clarity: The figures are well done and help to understand the concepts that are important to the comprehension of the paper. I liked that the contribution of the authors is well-defined and that the authors didn’t try to oversell their ideas. The authors should define the Q20 prediction score at line 229. On line 245, the author state "Although there are a number of issues with our approach …". I appreciate that the authors were honest about it but I would like them to elaborate on the said issues. Originality: I found the paper interesting and refreshing. I liked that the authors have identified a niche problem and that they addressed it simply. Significance: The authors were honest about their work being preliminary and still at an early stage, to which I agree. However, I believe that their work could have a significant impact in domains where data are in a spherical format or when 3D data is acquired from a fixed point sensor. I would like to emphasize that the impact of this paper will be greatly reduced if the author do not make the source code for the spherical convolution available online. In addition, having the source code to reproduce the results presented in the paper is good practice. For these reasons I strongly encourage the authors to make their source code available if the paper is accepted. Errors / typos: Line 68: "... based on the on the cubed …" Line 101: "... we can make the make the …" Line 117: The sentence seems unfinished or has a missing piece.